# ATNAS: Automatic Termination for Neural Architecture Search

## Abstract

Neural architecture search (NAS) is a framework for automating the design process of a neural network structure. While the recent one-shot approaches have reduced the search cost, there still exists an inherent trade-off between cost and performance. It is important to appropriately stop the search and further reduce the high cost of NAS. Meanwhile, the differentiable architecture search (DARTS), a typical one-shot approach, is known to suffer from overfitting. Heuristic early-stopping strategies have been proposed to overcome such performance degradation. In this paper, we propose a more versatile and principled early-stopping criterion on the basis of the evaluation of a gap between expectation values of generalisation errors of the previous and current search steps with respect to the architecture parameters. The stopping threshold is automatically determined at each search epoch without cost. In numerical experiments, we demonstrate the effectiveness of the proposed method. We stop the one-shot NAS algorithms and evaluate the acquired architectures on the benchmark datasets: NAS-Bench-201 and NATS-Bench. Our algorithm is shown to reduce the cost of the search process while maintaining a high performance.

## 1 Introduction

Deep learning (Goodfellow et al., 2016), the driving force of artificial intelligence which relies on stacked layers of operation units called neural networks, has made a paradigm shift from handcrafting good features to designing good neural architectures (combinations of operations) to extract good features. The success of such representation learning has been driven by improvements in neural architectures. However, limits to the heuristic designing of neural architectures remain. Neural architecture search (NAS) is designed to automate such architecture engineering of neural network models (Elsken et al., 2019; Ren et al., 2021). Early work on NAS adopted evolutionary computation (Angeline et al., 1994; Stanley & Miikkulainen, 2002; Floreano et al., 2008; Stanley et al., 2009; Jozefowicz et al., 2015), Bayesian optimisation (Bergstra et al., 2013; Domhan et al., 2015; Mendoza et al., 2016), and reinforcement learning (Baker et al., 2017; Zoph & Le, 2017; Zhong et al., 2018; Zoph et al., 2018) approaches to achieve state-of-the-art performance for various tasks. These methods, however, entail immense computational costs, requiring up to thousands of GPU days. Strubell et al. (2020) reported that a single NAS of the Transformer type (a large neural architecture) can produce as much carbon emission as five cars in their lifetime. The mainstream of NAS research is dedicated to reducing computational costs.

In the present research, we propose a stopping criterion for NAS. One can simply reduce extra search costs by stopping the search at an appropriate time. While early-stopping strategies have been proposed as a powerful solution to both reduce costs and improve performance (Baker et al., 2018; Li & Talwalkar, 2019), when to stop the search remains an important question to be addressed. We aim to early-stop NAS while maintaining its generalisation performance.

Our criterion particularly targets one-shot NAS, a prospective approach for reducing the cost of NAS, by customising the search strategy. One-shot NAS (Saxena & Verbeek, 2016; Brock et al., 2018; Pham et al., 2018; Bender et al., 2018) substantially reduces search costs by optimising a single redundant supernetwork, known as a one-shot model, representing all possible architectures in the search space as its sub-networks,

which share weights in common. More recent works combine this weight-sharing scheme with a continuous relaxation (Liu et al., 2019) or a stochastic relaxation (Xie et al., 2019; Akimoto et al., 2019) of the search space, which enables the use of gradient-based optimisation methods. These approaches, designated as differentiable NAS, have shown promising results while reducing the search costs. DARTS (Liu et al., 2019), a representative of differentiable NAS, has been successful at improving the search efficiency substantially and expanding the possibility of using NAS for more expensive tasks. Nevertheless, DARTS is well known to severely suffer from degenerated performance, which is sometimes worse than a random search (Yu et al., 2020; Yang et al., 2020). Parameter-less operations such as skip connections have been observed to dominate the generated architecture (Liang et al., 2019; Chu et al., 2020; Zela et al., 2020). Wide and shallow structures are preferred by DARTS (Shu et al., 2020). It has been demonstrated that failures of DARTS have been attributed to the discretisation of the architecture parameters fitted in the sharp minimum (Zela et al., 2020).

RobustDARTS (Zela et al., 2020) mitigates this degeneration of performance by monitoring the dominant eigenvalue of the Hessian matrix of validation loss w.r.t. the architectural parameters, which is correlated with the sharpness of the loss landscape. The search should be stopped before the eigenvalue increases excessively so that flat minima and the generalisation are expected. According to Liang et al. (2019), the over-parameterised weights of the model are overfitted to the training data and the less-parameterised architecture parameters are underfitted to the validation data, resulting in a gap between the training and validation errors. To prevent this, Liang et al. (2019) proposed DARTS+ as an early-stopping scheme.

Both Zela et al. (2020) and Liang et al. (2019) performed an empirical study to determine a handcrafted stopping criterion. RobustDARTS uses a fixed value of the ratio of the smoothed maximum eigenvalue of the Hessian as the threshold. This heuristic criterion is underpinned by an empirical study on the relationship between the generalisation performance and the Hessian eigenvalues and said to have been worked well for several types of tasks and search spaces; however, the versatility among other search spaces and other types of tasks is still unknown. Moreover, Hessian/eigenvalue computation has a high cost.

DARTS+ stops the search in DARTS when the number of skip-connections becomes more than two or the ranking of the architecture parameters becomes stable. The former criterion is motivated by P-DARTS (Chen et al., 2019), which limits the number of skip-connects in the cell or the unit structure to be optimised of the final architecture to two by manually cutting down and is simple and easy to implement. The latter criterion considers the stability of the ranking of the architecture parameters over the manually determined number of epochs. A better stopping criterion based on the selection of operations has recently been proposed by Zhang & Ding (2021). However, both of ad hoc criteria may not be versatile among other search spaces and may have no theoretical guarantee of generalisation performance.

ASNG-NAS (Akimoto et al., 2019) is a yet another paradigm of gradient-based one-shot NAS. In this method, the expected value of the generalisation error w.r.t. the distribution of architecture parameters is optimised. Whereas a sampling-based exploration method such as ASNG-NAS is promising because its performance is superior to DARTS with a lower GPU cost, automatic termination for such a sampling-based method has not yet been considered to this date. In this paper, on the basis of the formulation of ASNG-NAS, we propose a stopping criterion called Automatic Termination for Neural Architecture Search (ATNAS). Unlike a related study (Makarova et al., 2022), which considers early-stopping of BO-based NAS, or DARTS+ and RobustDARTS, which are only suitable for DARTS, our method is not limited to any search paradigms.

Ideally, the search should be stopped by monitoring the generalisation/test performance because we want a model with a smaller error for unknown data. During the search, generalisation errors are not directly trackable. Instead, we consider a gap between the expectation values of generalisation errors of the previous and current search steps w.r.t. the architecture parameters, and we evaluate and adopt the upper bound of it as the stopping criterion. When the upper bound becomes less than the estimated standard deviation of the generalisation error, the architecture search is terminated because we cannot expect further significant reduction of the generalisation error.

Finally, the trade-off between cost and performance still remains a challenge for NAS. ASNG-NAS is extremely light; however, in a scenario such as multi-objective NAS, in which many trials are required to adjust the factor of the penalty term (such as FLOPS) on the objective function, efficiency is demanded. In such

cases, early-stopping is essential from the point of view of reducing energy consumption as well as improving engineering efficiency. Moreover, appropriate termination sometimes contributes to improved performance, as suggested by our experimental results. The proposed method is essentially parallel to other acceleration and stabilisation methods, and it is expected to be effective in principle to combine them with the proposed method to further increase efficiency. Other methodologies that reduce the computational cost only optimise the architecture for validation data and do not consider generalisation errors. They do not necessarily give good performance when evaluated on a test-set (after re-training).

The contributions of this study are summarised as follows:

- We propose a stopping criterion for neural architecture search based on a gap between expectation values of generalisation errors of the previous and current search steps w.r.t. the architecture parameters. The stopping threshold is automatically determined at each search epoch. The proposed method can reduce computational costs while considering generalisation errors.

- We conduct numerical experiments to demonstrate that the proposed method can early-stop the search while maintaining the generalisation performance. We stop the one-shot NAS algorithms such as ASNG-NAS and DARTS and evaluate the acquired architectures using benchmark datasets.

The rest of this paper is organised as follows. Section 2 describes the preliminaries. Section 3 summarises the existing algorithms our method is based on. Section 4 defines our stopping criterion. Section 5 demonstrates the effectiveness of the proposed method. Section 6 mentions some limitations of our approach. Section 7 presents our conclusions.

## 2 Preliminaries and problem setting

In this section, we briefly introduce the NAS framework (Elsken et al., 2019; Wistuba et al., 2019). We consider learning a predictive model of the form $f_{\boldsymbol{w},\boldsymbol{c}} : \mathcal{X} \to \mathcal{Y}$, which maps input data $\boldsymbol{x} \in \mathcal{X}$ to its corresponding output $y \in \mathcal{Y}$, where $\mathcal{X}$ and $\mathcal{Y}$ are input and output spaces, respectively. The predictive model has two kinds of parameters: $\boldsymbol{w} \in \mathcal{W}$ is a parameter and $\boldsymbol{c}$ is a hyperparameter. Typically, $\boldsymbol{c}$ determines the network architecture and $\boldsymbol{w}$ is the connection weight of a fixed network specified by $\boldsymbol{c}$.

The optimisation goal is to find a set of parameters $\boldsymbol{w} \in \mathcal{W}$ and hyperparameters $\boldsymbol{c} \in \mathcal{C}$ that minimise the generalisation error $\mathcal{L}$ of the neural network as a predictive model. For a loss function $\ell$ and the true data distribution $\mathcal{D}$ generating a sample $S_n = \{(\boldsymbol{x}_i, y_i)\}_{i=1}^n$, we respectively define the generalisation error and empirical error as

$$\mathcal{L}_{\mathcal{D}}(\boldsymbol{w}, \boldsymbol{c}) := \mathbb{E}_{(\boldsymbol{x},y)\sim\mathcal{D}}\left[\ell(y, f_{\boldsymbol{w},\boldsymbol{c}}(\boldsymbol{x}))\right], \tag{1}$$

and

$$\mathcal{L}_{S_n}(\boldsymbol{w}, \boldsymbol{c}) := \frac{1}{|S_n|} \sum_{(\boldsymbol{x}_i, y_i) \in S_n} \ell(y_i, f_{\boldsymbol{w},\boldsymbol{c}}(\boldsymbol{x}_i)) \tag{2}$$

where $|S|$ is the size of the dataset $S$. We consider that the training, validation, and test datasets are i.i.d. samples from $\mathcal{D}$, which are denoted by $S_{\text{train}} = \{(\boldsymbol{x}_i, y_i)\}_{i=1}^{n_{\text{train}}}$, $S_{\text{val}} = \{(\boldsymbol{x}_i, y_i)\}_{i=1}^{n_{\text{val}}}$, and $S_{\text{test}} = \{(\boldsymbol{x}_i, y_i)\}_{i=1}^{n_{\text{test}}}$, respectively.

NAS solves two nested problems: weight optimisation $\underset{\boldsymbol{w}\in\mathcal{W}}{\text{minimise}}\ \mathcal{L}_{S_{\text{train}}}$ and architecture optimisation $\underset{\boldsymbol{c}\in\mathcal{C}}{\text{minimise}}\ \mathcal{L}_{S_{\text{val}}}$. Recent NAS approaches adopt a weight sharing strategy in which the architecture search space $\mathcal{C}$ is encoded in a supernet, $f(\boldsymbol{w}, \boldsymbol{c})$. Here we note that $f(\boldsymbol{w}, \boldsymbol{c})$ is a synonym of $f_{\boldsymbol{w},\boldsymbol{c}}(\boldsymbol{x})$. We hereafter focus on the optimisation of $\boldsymbol{w}$ and $\boldsymbol{c}$; thus, we omit $\boldsymbol{x}$ and use this notation to make $\boldsymbol{w}$ and $\boldsymbol{c}$ explicitly the targets of optimisation. The supernet is trained once and, as subgraphs of the supernet, all architectures inherit their weights directly from $\boldsymbol{w}$. Typical weight sharing approaches convert the discrete architecture search space into a continuous space, i.e., $\boldsymbol{c}$ is re-parameterised as $\boldsymbol{c}(\theta)$ by a real-valued vector $\theta$. Both the weights and the architecture parameter $\theta$ are trained via a bi-level optimisation as $\theta^* = \underset{\theta}{\arg\min}\ \mathcal{L}_{S_{\text{val}}}(\boldsymbol{w}_\theta^*, \boldsymbol{c}(\theta))$ s.t. $\boldsymbol{w}_\theta^* = \underset{\boldsymbol{w}}{\arg\min}\ \mathcal{L}_{S_{\text{train}}}(\boldsymbol{w}, \boldsymbol{c}(\theta))$ .

## 3 Related works

In this section, we overview the related works on which our methodology is based.

### 3.1 Stopping methods for other machine learning problems

Early stopping of learning is an important issue, and there are several studies in contexts other than NAS. Hyperparameter optimisation is usually structured as an "outer-loop" for hyperparameter searches, and an "inner-loop" for optimisation of the parameters of the predictive model with the hyperparameters. Methods for early stopping of the learning of predictive model parameters in the inner-loop, such as the connection weights of a neural network, have been studied for a long time (Prechelt, 1996). In the context of Bayesian optimisation, several methods have been proposed to increase the overall efficiency by scheduling the search while stopping the inner-loop in each hyper-parameter early (Swersky et al., 2014; Li et al., 2017; Falkner et al., 2018; Dai et al., 2019). In contrast, there are few methods to appropriately stop the outer-loop. Some of them are specific to acquisition functions (Lorenz et al., 2015; Nguyen et al., 2017), such as the probability of improvement (Kushner, 1964) and expected improvement (Mockus et al., 1978). There is a recently proposed method which is more general with an automatic thresholding mechanism (Makarova et al., 2022), which is also referred to in this paper. In addition, several optimal stopping methods for active learning (Settles, 2009) have been studied, ranging from heuristic methods (Bloodgood & Vijay-Shanker, 2009; Altschuler & Bloodgood, 2019; Kurlandski & Bloodgood, 2022) to that based on learning theory (Ishibashi & Hino, 2020), but the problem is different from hyper-parameter searches such as NAS.

There are only few criteria for optimal stopping methods equipped with NAS algorithms. We describe these methods in some detail hereinafter.

### 3.2 DARTS

The search space of DARTS (Liu et al., 2019) is represented as cells defined by a directed acyclic graph (DAG) of $N$-nodes. In the DAG, a node $i \in 0, \ldots, N-1$ is a latent expression similarly to a feature map in the convolution architecture. The edge $(i,j)(i < j)$ is a flow or a transformation between feature maps from a node $i$ to a node $j$. A set $\mathcal{O}$ of $K$ operations is expressed as $\mathcal{O} = \{o_0, o_1, \ldots, o_{K-1}\}$ and each operation $o_\nu(\nu \in \{1, \ldots, K-1\})$ includes trainable weights $w_\nu^{(i,j)}$ such as weights of convolution. A set of operations $\mathcal{O}$ and its elements are defined in common for all edges $(i,j)$, while the weights $\boldsymbol{w}_{(i,j)}$ used in the operations are provided separately for each edge. The architecture parameter $\alpha_\nu^{(i,j)}$ is the appropriateness of the $\nu$-th operation at the edge between the node $i$ and the node $j$ and introduced as the weight after applying the operation $o_\nu^{(i,j)}$. For $i < j$, the calculation from the node $i$ to the node $j$ is expressed as $\sum_{\nu=0}^{K-1} \frac{\exp(\alpha_\nu^{(i,j)})}{\sum_{\nu'=0}^{K-1} \exp(\alpha_{\nu'}^{(i,j)})} \cdot o_\nu(x_i)$.

**Stopping criterion: DARTS+** Liang et al. propose two stopping criteria for DARTS (Liang et al., 2019). One of the criteria explicitly limits the number of skip-connects. The search process is terminated whenever there are two or more skip-connections in one normal cell. Another criterion monitors the ranking of architecture parameters $\alpha$ for learnable operations and when it becomes stable for 10 epochs the search is terminated, since the stable ranking of the architecture parameter $\alpha$ for the learnable operations indicates a saturated search procedure in DARTS.

**Stopping criterion: RobustDARTS** Zela et al. have carried out exploratory research to investigate the correlation between the generalisation performance and the dominant eigenvalue of the Hessian of the validation loss w.r.t. the architectural parameters over search time (Zela et al., 2020). The large eigenvalue of the Hessian correlates with the performance degradation. In order to avoid the large dominant eigenvalue serving as a proxy for the sharpness of loss landscape, they propose to stop search when $\bar{\lambda}_{\max}^\alpha(i-k)/\bar{\lambda}_{\max}^\alpha(i) < 0.75$. Here, $\bar{\lambda}_{\max}^\alpha(i)$ is the dominant eigenvalue of the Hessian of validation loss w.r.t. the parameters averaged over $k = 5$ epochs around $i$ and return the architecture at $i - k$.

### 3.3 ASNG-NAS

In ASNG-NAS (Akimoto et al., 2019), optimisation of the expected value of the generalisation error $\mathcal{L}$ with respect to $p_\theta$, where $p_\theta$ is a distribution on the categorical space $\mathcal{C}$, is considered as

$$J(\boldsymbol{w}, \theta) := \int_{\boldsymbol{c} \in \mathcal{C}} \mathcal{L}(\boldsymbol{w}, \boldsymbol{c}) p_\theta(\boldsymbol{c}) d\boldsymbol{c} = \mathbb{E}_{p_\theta}[\mathcal{L}(\boldsymbol{w}, \boldsymbol{c})], \tag{3}$$

where $\boldsymbol{w}$, $\boldsymbol{c}$, $\mathcal{L}$ represent connection weights, architecture parameters, and the generalisation error, respectively. The architecture parameter $\boldsymbol{c} \sim p_\theta(\boldsymbol{c})$ is a one-hot vector sampled from a categorical distribution $p_\theta(\boldsymbol{c})$, which determines the neural architecture.

The single search iteration of ASNG-NAS includes: (1) The sampling of an architecture: $\boldsymbol{c} \sim p_\theta(\boldsymbol{c})$, (2) The update of $\boldsymbol{w}$: $\boldsymbol{w}_{t+1} = \boldsymbol{w}_t + \eta_{\boldsymbol{w}} \nabla_{\boldsymbol{w}} J_\theta(\boldsymbol{w}_t, \theta_t)$, (3) The update of $\theta$: $\theta_{t+1} = \theta_t + \eta_\theta^t \tilde{\nabla}_\theta J_\theta(\boldsymbol{w}_{t+1}, \theta_t)$, and (4) The adaptation of step-size, where $\tilde{\nabla}_\theta = F(\theta_t)^{-1} \nabla_\theta$ is the natural gradient (Amari, 1998). Note that the gradients are approximated by Monte-Carlo method in practice.

**Convergence of $\theta$** The original paper of ASNG-NAS does not automatically stop the search; instead, the search runs for the fixed epochs, e.g., 100 epochs. The convergence of $\theta$, $\frac{1}{d} \sum_{i=1}^{d} \max_j [\theta]_{i,j}$, where $d$ is the cell index and $j$ is the operation index, is monitored. It is the average value of the probability vector $\theta$ of the categorical distribution which converges to a certain category. In the paper by Akimoto et al. (2019), it is argued that it reaches 0.9 in about 50 epochs and thus the convergence is fast. Nevertheless it is not trivial to use this convergence of $\theta$ as a stopping criterion. In the supplementary material, we compare our metric (the upper bound of the gap of the expected generalisation errors) with the convergence of $\theta$ and discuss that our metric reflects the generalisation performance better than the convergence of $\theta$.

## 4 Proposed method

This section presents the Automatic Termination for Neural Architecture Search (ATNAS), a stopping criterion based on the evaluation of the gap between expected generalisation errors with previous and current hyperparameter distributions.

### 4.1 Upper bound for gap between expected values of generalisation errors

The aim of NAS is to find the neural architecture which minimises the generalisation error defined in equation 1. So, it is natural to focus on the gap between the generalisation errors with hyperparameter distributions searched at previous and current step. Adopting the stochastic relaxation, neural architecture is sampled from distribution $p_\theta(\boldsymbol{c})$. We consider the gap of the expected values of generalisation errors with respect to $p_{\theta_t}(\boldsymbol{c})$ and $p_{\theta_{t-1}}(\boldsymbol{c})$:

$$\Delta \mathcal{L} = J(\boldsymbol{w}_t, \theta_t) - J(\boldsymbol{w}_{t-1}, \theta_{t-1}) = \mathbb{E}_{p_{\theta_t}} [\mathcal{L}(\boldsymbol{w}_t, \boldsymbol{c})] - \mathbb{E}_{p_{\theta_{t-1}}} [\mathcal{L}(\boldsymbol{w}_{t-1}, \boldsymbol{c})],$$

which is the gap between the objective function of ASNG-NAS, equation 3, hence expected to converge to zero when search is sufficiently progressed.

The gap between the generalisation errors with hyperparameter distributions searched at previous and current step is not directly computable, because the generalisation error is not available during the architecture search. Our strategy is to estimate the upper bound of the gap $\Delta \mathcal{L}$ and determine the termination timing of the search process by comparing the upper bound with the threshold. So as to evaluate the upper bound $r_t \geq \Delta \mathcal{L}$ of the gap, we consider the Pinsker's inequality below (Theorem 4.1).

**Theorem 4.1** ((Russo & Roy, 2016), Fact 9)**.** *Let $p_{\theta_t}(\boldsymbol{c})$ and $p_{\theta_{t-1}}(\boldsymbol{c})$ be arbitrary probability distributions of a vector valued random variable $\boldsymbol{c}$. The following inequality holds for an arbitrary measurable function $\mathcal{L} \in [a, b]$*

$$\mathbb{E}_{p_{\theta_t}(\boldsymbol{c})}[\mathcal{L}(\boldsymbol{w}, \boldsymbol{c})] - \mathbb{E}_{p_{\theta_{t-1}}(\boldsymbol{c})}[\mathcal{L}(\boldsymbol{w}, \boldsymbol{c})] \leq (b-a)\sqrt{\frac{1}{2} D_{\mathrm{KL}}(p_{\theta_t}(\boldsymbol{c}) \| p_{\theta_{t-1}}(\boldsymbol{c}))},$$

*where a, b are the maximum and minimum values of L, and $D_{\mathrm{KL}}(p\|q)$ is the Kullback–Leibler (KL) divergence between distributions p and q.*

We note that $p_\theta(\boldsymbol{c})$ is a discrete distribution in ASNG-NAS, hence it is easy to calculate the KL divergence.

We can calculate the upper bound via the above theorem when $\boldsymbol{w}_{t+1} = \boldsymbol{w}_t$. However, the weights $\boldsymbol{w}$ are updated in the NAS procedure; hence, the theorem (4.1) is not directly applicable. We approximate it via Taylor series expansion.

Let $n$-th Taylor expansion of $J(\boldsymbol{w},\theta)$ around $\boldsymbol{w}_t$ with respect to $\boldsymbol{w}$ as $T_n(\boldsymbol{w},\theta)$, and its reminder term as $R_n(\boldsymbol{w}) := J(\boldsymbol{w},\theta) - T_n(\boldsymbol{w},\theta)$. We define $T'_n(\boldsymbol{w},\theta)$ as $T_n(\boldsymbol{w},\theta)$ without the 0-th term. That is, $T'_n(\boldsymbol{w}_t,\theta_t) = T_n(\boldsymbol{w}_t,\theta_t) - T_0(\boldsymbol{w}_t,\theta_t) = T_n(\boldsymbol{w}_t,\theta_t) - J(\boldsymbol{w}_{t-1},\theta_t)$. Then, we have

$$J(\boldsymbol{w}_t,\theta_t) - J(\boldsymbol{w}_{t-1},\theta_{t-1}) = T_n(\boldsymbol{w}_t,\theta_t) + R_n(\boldsymbol{w}_t,\theta_t) - J(\boldsymbol{w}_{t-1},\theta_{t-1}) \tag{4}$$

$$= J(\boldsymbol{w}_{t-1},\theta_t) - J(\boldsymbol{w}_{t-1},\theta_{t-1}) + T'_n(\boldsymbol{w}_t,\theta_t) + R_n(\boldsymbol{w}_t,\theta_t). \tag{5}$$

Therefore, we apply the theorem 4.1 to the 1st and the 2nd term of equation 5 to obtain the following inequality:

$$J(\boldsymbol{w}_t,\theta_t) - J(\boldsymbol{w}_{t-1},\theta_{t-1}) \leq (b-a)\sqrt{\frac{1}{2}D_{\mathrm{KL}}(p_{\theta_t}(\boldsymbol{c})\|p_{\theta_{t-1}}(\boldsymbol{c}))} + T'_n(\boldsymbol{w}_t,\theta_t) + R_n(\boldsymbol{w}_t,\theta_t).$$

We then remove the residual term $R_n(\boldsymbol{w},\theta)$. Let $J$ as $C^n$-class with respect to $\boldsymbol{w}$. Assume that $J(\boldsymbol{w},\theta) \leq M$ for all $\boldsymbol{w} \in [\boldsymbol{w}_{t-1}, \boldsymbol{w}_{t-1} + d]$ with $\|\boldsymbol{w}_t - \boldsymbol{w}_{t-1}\| < d$. Then we obtain the following inequality:

$$J(\boldsymbol{w}_t,\theta_t) - J(\boldsymbol{w}_{t-1},\theta_{t-1}) \leq (b-a)\sqrt{\frac{1}{2}D_{\mathrm{KL}}(p_{\theta_t}(\boldsymbol{c})\|p_{\theta_{t-1}}(\boldsymbol{c}))} + T'_n(\boldsymbol{w}_t,\theta_t) + \frac{M}{n!}\|\boldsymbol{w}_t - \boldsymbol{w}_{t-1}\|^n.$$

$T'_n$ is the $n$-th degree Taylor polynomials without the 0-th term and $J(\boldsymbol{w},\theta) \leq M$.

Assuming that the range of the generalisation error $\mathcal{L}$ is $\mathcal{L} \in [a,b]$, we consider $\hat{\mathcal{L}} := \mathcal{L} - a$. In this case, the difference between expectation values of $\hat{\mathcal{L}}$ are the same as that of expected values of $\hat{\mathcal{L}}$, so the upper bound remains unchanged. We consider the upper bound of $\hat{\mathcal{L}}$. Because $\hat{\mathcal{L}} \in [0, b-a]$, defining $M = (b-a)$, we can use the Taylor's inequality. Concrete forms of the upper bound $r_t$ of $\Delta\mathcal{L}$ with 1st and 2nd order Taylor expansion are given as

$$r_t^1 = (b-a)\sqrt{\frac{1}{2}D_{\mathrm{KL}}(p_{\theta_t}(\boldsymbol{c})\|p_{\theta_{t-1}}(\boldsymbol{c}))} + \nabla_{\boldsymbol{w}}J(\boldsymbol{w}_{t-1},\theta_t) \cdot (\boldsymbol{w}_t - \boldsymbol{w}_{t-1})$$
$$+ (b-a)\|\boldsymbol{w}_t - \boldsymbol{w}_{t-1}\|, \tag{6}$$

$$r_t^2 = (b-a)\sqrt{\frac{1}{2}D_{\mathrm{KL}}(p_{\theta_t}(\boldsymbol{c})\|p_{\theta_{t-1}}(\boldsymbol{c}))} + \nabla_{\boldsymbol{w}}J(\boldsymbol{w}_{t-1},\theta_t) \cdot (\boldsymbol{w}_t - \boldsymbol{w}_{t-1})$$
$$+ \frac{1}{2}(\boldsymbol{w}_t - \boldsymbol{w}_{t-1})^\mathsf{T}\nabla_{\boldsymbol{w}}^2 J(\boldsymbol{w}_{t-1},\theta_t)(\boldsymbol{w}_t - \boldsymbol{w}_{t-1}) + \frac{(b-a)}{2}\|\boldsymbol{w}_t - \boldsymbol{w}_{t-1}\|^2. \tag{7}$$

In general, we do not have access to the range of $\mathcal{L}$ nor the range of loss function is unbounded. One possibility is to use the bounded loss function such as a bi-tempered logistic loss (Amid et al., 2019), which is a common loss function for practitioners to deal with noisy labels (see supplementary material for details). Another practical approach is assuming the range of a loss function. From our experimental results, we see that both approaches work well.

## 4.2 Automatic determination of threshold

We propose to stop architecture search when the upper bound $r_t$ becomes less than the threshold $\lambda_t$. In our setting, or a typical NAS, validation data are available and it motivates us to auto-tune the termination threshold using training/validation data. In a recent work (Makarova et al., 2022), the early stopping

criterion is presented to mitigate the issue of overfitting in tuning hyperparameters of predictive models by Bayesian optimisation. The termination is triggered once the upper bound of a simple regret becomes less than the standard deviation of the generalisation error, which can be estimated from $k$-fold cross validation of training and validation data as

$$\lambda_t = \sqrt{\frac{1}{k} + \frac{|S_{\text{val}}|}{|S_{\text{train}}|}} \, \hat{s}_t, \tag{8}$$

where $\hat{s}_t^2$ is the estimated sample variance of $k$-fold cross-validation (Nadeau & Bengio, 1999) w.r.t. the predictive model at the $t$-th epoch of NAS. In the ASNG-NAS case, $\hat{s}_t^2$ is calculated by, instead of performing cross-validation, we split the validation data (5-fold) and calculate the standard deviation of validation errors, with the hyperparameter $c$ independently sampled from $p_\theta$ at each epoch. Our proposed criterion is based on the gap between (expected) generalisation errors, and it is natural to compare it with the estimated standard deviation of the generalisation error. Here we note that the estimated standard deviation $\lambda_t$ is easy to compute for each epoch of architecture search in ASNG-NAS without undue overhead. We benchmark the computational time required for the each stopping algorithm in the supplementary material.

### 4.3 Application to DARTS

The proposed method uses the ASNG-NAS setting for formulating a stopping criterion and hence is not directly applicable to DARTS. In particular, in DARTS, the optimisation target is not the expected value of the generalisation error, and the architecture parameter is not based on the stochastic relaxation but the continuous relaxation. We interpret the softmax $\frac{\exp(\alpha_\nu^{(i,j)})}{\sum_{\nu'=0}^{K-1} \exp(\alpha_{\nu'}^{(i,j)})}$ as the multinomial parameter distribution and compare it with the distribution at the previous search epoch. The $p_t$ and $p_{t-1}$ in the definition of $r_t$ that we use as a stopping criterion are the posterior distributions calculated by the softmax. For the automatic determination of the threshold, a one-hot discretisation is performed and the the threshold is automatically calculated by the equation 8 at the end of each epoch.

## 5 Experiments

To evaluate the efficiency of ATNAS, we conducted numerical experiments[1]. The architecture search is carried out by DARTS or ASNG-NAS. For the stopping criterion, we evaluate the upper bound via equation 6. The termination threshold is calculated by equation 8. For the loss function, we adopt the bi-tempered logistic loss, of which the upper bound is defined, to satisfy the assumptions of the theorem. We also investigate the commonly used cross-entropy loss. We evaluate the results on the basis of search cost (number of epochs) and test accuracy.

### 5.1 Dataset

We use benchmark datasets including NAS-Bench-201 database (Dong & Yang, 2020) and NATS-Bench database (Dong et al., 2021) constructed on CIFAR-10/CIFAR-100 (Krizhevsky, 2009) and ImageNet (Deng et al., 2009), and adopt the standard preprocessing and data augmentation following the previous works.

### 5.2 Implementation details

We use the original implementations of DARTS, ASNG-NAS, NAS-Bench-201 and NATS-Bench. We use the bi-tempered logistic loss or the cross entropy loss for the loss function. Experiments are performed on a single NVIDIA QUADRO RTX 6000 GPU. For the cross entropy loss, we assume the upper bound $b$ in Theorem 4.1 to be the loss value of the first epoch, and the lower bound $a$ to be zero. For the stopping criterion, we adopt the 1st order expansion given by equation 6. We note that the 2nd order expansion offers tighter upper bound, but calculation of Hessian w.r.t. parameter $w$ requires heavy computation burden.

---

[1]The source code can be found in the supplementary material.

## 5.3 Results

All of the experiments are repeated for five trials and the mean values and standard deviations are reported.

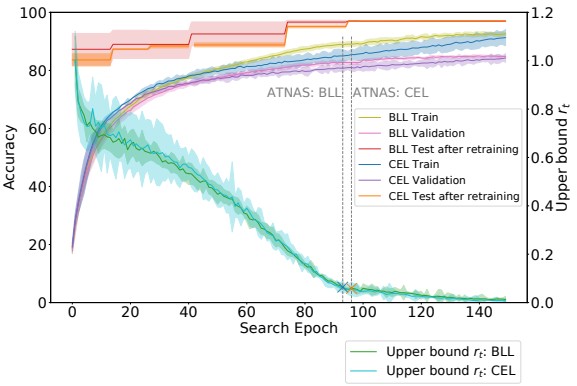

Figure 1: Automatic termination of ASNG-NAS: CIFAR-10 accuracy curves vs. upper bound $r_t$. Red dashed lines represent one terminated timing by ATNAS. BLL: Bi-tempered logistic loss, CEL: Cross entropy loss.

Figure 2: Dominant eigenvalues of the Hessian w.r.t. the parameters (blue), the proposed upper bound $r_t$ of the gap between generalisation errors (purple), and the CIFAR-10 test accuracy obtained by querying NAS-Bench-201 (olive).

### 5.3.1 Automatic termination of ASNG-NAS

We conduct experiments to establish whether the proposed method can reasonably halt ASNG-NAS. Figure 1 shows the training/validation/test accuracy of ASNG-NAS for CIFAR-10, and the stop timing determined by ATNAS. The bi-tempered logistic loss and the cross-entropy loss are used. We confirm that there is no significant difference between the cases for the bi-tempered logistic loss and the cross entropy loss. The solid lines are the mean values, and the shaded areas are the standard deviations. The blue cross mark and red vertical line represent the terminated timing for one trial (note that for the five trials, there is only minor difference in the determined stop time, hence we only show the stop time for the first trial). While the test accuracies appear to continue improving slightly, the improvement is considered to be saturated and hence the further search is questionable from the cost-effectiveness point of view. Thus, the determined termination is at a reasonable timing. A comparison of the proposed criteria and the convergence of $\theta$ is shown in the supplementary material.

### 5.3.2 Comparison with Hessian considered in RobustDARTS

We consider terminating DARTS with the proposed criterion and RobustDARTS. We train DARTS with CIFAR-10 and compare the dominant eigenvalues of Hessian of the validation loss, upper bound $r_t$, and the test accuracy values by querying NAS-Bench-201. The early stopping method equipped with RobustDARTS monitors the Hessian eigenvalue which does not sometimes capture the performance degradation and might fail to stop the search at the right time, as shown in Figure 2. The eigenvalues of the Hessian continue to be lower in the early stages until when the performance degradation begins. The eigenvalues increase rapidly after the degradation; nevertheless their response is slow and unstable. In contrast, the upper bound $r_t$ as our stopping criterion is continuously decreasing and with the appropriate termination threshold the search can be terminated in the early stage before the performance degradation happens.

### 5.3.3 Automatic termination of DARTS

Early-stopping strategies proposed for DARTS include DARTS+ and RobustDARTS. We compare our method with these conventional methods by the stopping time and the accuracy of the obtained architectures. ATNAS on DARTS is evaluated using NAS benchmarks including NAS-Bench-201 and NATS-Bench.

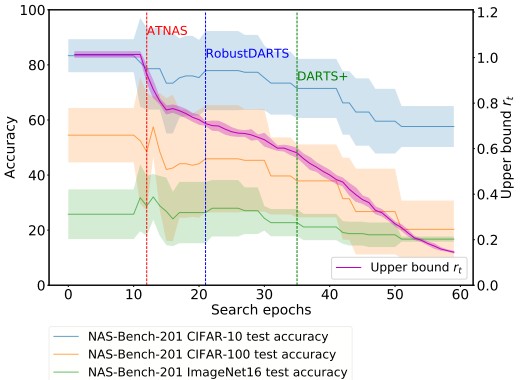

(a) DARTS with bi-tempered logistic loss trained on CIFAR-10. The baseline accuracy values are obtained by querying NAS-Bench-201.

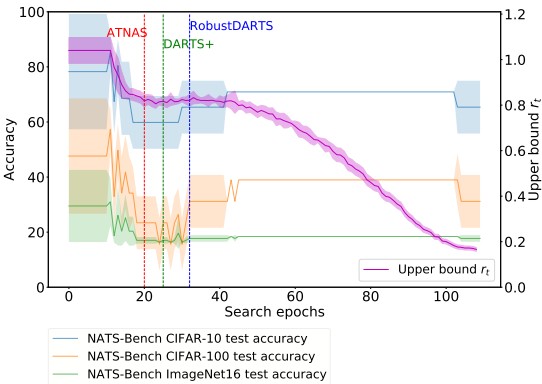

(b) DARTS with bi-tempered logistic loss trained on CIFAR-10. The baseline accuracy values are obtained by querying NATS-Bench.

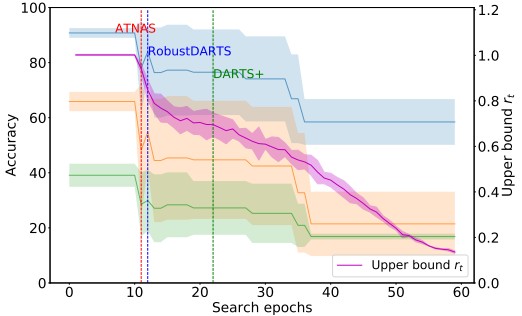

(c) DARTS with cross entropy loss trained on CIFAR-10. The baseline accuracy values are obtained by querying NAS-Bench-201.

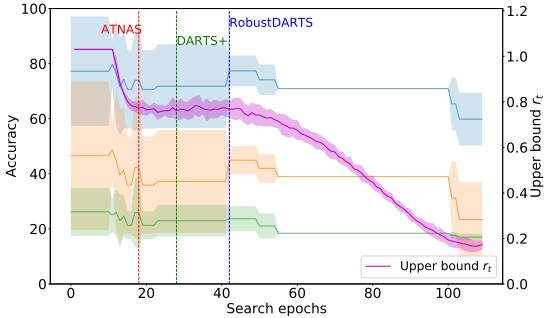

(d) DARTS with cross entropy loss trained on CIFAR-10. The baseline accuracy values are obtained by querying NATS-Bench.

Figure 3: Automatic termination of DARTS by conventional and proposed criteria.

We employ the warm-up scheme which is widely used in supernet training. The parameters are frozen at the beginning for 10 epochs while updating weights. The accuracy of the obtained architectures and the quantity used for our criterion are shown in Figure 3. The results are averaged for 5 trials and the solid lines are the mean values and the shaded areas are the standard deviations. The red, blue, and green dashed vertical lines represent the terminated timings by ATNAS, RobustDARTS, and DARTS+, respectively, for one trial. By observation, we note that the results of the bi-tempered logistic loss and the cross entropy loss with the empirical upper bound are essentially the same and, thus, our stopping criterion is compatible with the standard NAS with the widely used cross-entropy loss.

While DARTS performs worse than a random search on benchmarks, ATNAS (along with other early-stopping methods) is able to stop the search before the severe performance degradation occurs. Our early-stopping method terminates the search surprisingly early. The threshold is automatically determined by estimating the standard deviation of the generalisation error via the standard deviation of the 5-fold validation error. In the case where the variation of the validation error is large, the threshold value becomes high enough to stop the search early. Particularly in the case of the DARTS, it implies that redundant search time has been previously spent. The proposed method is not only avoiding the performance degradation but also cost effective and environmentally friendly. Existing approaches are likely to attempt to stop for curves after discretised. It is shown that failures of DARTS are mostly due to the discretisation (Zela et al., 2020; Wang et al., 2021) and it is reasonable to terminate the search by monitoring the curves before discretised which the proposed method focuses on.

# 6 Limitations

The intrinsic limitation of our method is that it requires the range of loss function to be bounded and known. We can use a popular bounded loss, and practically, the cross-entropy loss with an empirical bound is effective, but removing this restriction is an important problem to be solved.

The computational complexity for calculating our termination metric with the 1st order Taylor expansion is $O(N + M)$, where $N$ is the number of network parameters and $M$ is the dimension of the architecture parameters. This extra computational cost is negligible compared to those required by a standard NAS procedure. However, when we want to improve the tightness of the upper bound by computing higher order Taylor expansion, the cost would matter.

# 7 Conclusion and future direction

In this study, we proposed a stopping criterion for one-shot neural architecture search. Stopping NAS at the appropriate time is very important for both Green-AI (Strubell et al., 2020; Schwartz et al., 2020) and AutoML (He et al., 2021; Santu et al., 2022). The proposed criterion monitors the gap between expectation of generalisation errors of the previous and current search steps with respect to the architecture parameters. It is not directly computable because we do not have access to the generalisation error. To address this problem, we derived an easy to compute upper bound. By way of experimentation, we demonstrated that the proposed criterion is useful to terminate differential architecture searches of both types appropriately: stochastic relaxation (ASNG-NAS) and continuous relaxation (DARTS). The architecture search is terminated with high performance. While the conventional methods are limited to specific search spaces, the proposed method is versatile and widely applicable to one-shot models. There are recent researches on zero-shot approach (Mellor et al., 2020; Abdelfattah et al., 2021; Chen et al., 2021; Lin et al., 2021; Ning et al., 2021) and extrapolation approach (Swersky et al., 2014; Domhan et al., 2015; Klein et al., 2017; Baker et al., 2018; Rawal et al., 2019; Wistuba & Pedapati, 2020; Yan et al., 2021) to speed up NAS and it is promising to combine these methods with our ATNAS to further improve the efficiency of NAS.

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

# A  Appendix

## A.1  Upper bound for Bi-Tempered Logistic Loss

One example of the bounded loss function is a Bi-Tempered Logistic Loss which adopts parameters to adjust the maximum of the margins and the damping ratio of the SoftMax function (Amid et al., 2019):

$$\forall 0 \leqslant t_1 < 1 < t_2 : L_{t_1}^{t_2}(\hat{\boldsymbol{a}} \mid \boldsymbol{y}) := \Delta_{F_{t_1}}(\boldsymbol{a} - \xi_{t_2}(\hat{\boldsymbol{a}})), \text{with } \xi_t \text{ s.t. } \sum_{i=1}^{k} \exp_t (a_i - \xi_t(\boldsymbol{a})) = 1. \tag{9}$$

For each input $\boldsymbol{x}$, the input to the SoftMax layer is $\boldsymbol{z}$ and the activation function as linear transformation for each class $i$ is defined as $\hat{a}_i = \boldsymbol{w}_i \cdot \boldsymbol{z}$. $\Delta_{F_t}$ is a Bregman divergence defined by a convex function $F_t$.

$$\Delta_{F_t}(\boldsymbol{y}, \hat{\boldsymbol{y}}) = \sum_{i=1}^{k} \left( y_i \log_t y_i - y_i \log_t \hat{y}_i - \frac{1}{2-t} y_i^{2-t} + \frac{1}{2-t} \hat{y}_i^{2-t} \right)$$

$$= \sum_{i=1}^{k} \left( \frac{1}{(1-t)(2-t)} y_i^{2-t} - \frac{1}{1-t} y_i \hat{y}_i^{1-t} + \frac{1}{2-t} \hat{y}_i^{2-t} \right). \tag{10}$$

$\exp_t$ is the inverse function of $\kappa$-deformed logarithm $\log_t(x) := \frac{1}{1-t}(x^{1-t} - 1)$:
$\exp_t(x) := [1 + (1-t)x]_+^{1/1-t}$.

The upper bound of $\Delta_{F_t}(y, \hat{y})$ is given as follows (Amid et al., 2019):

$$\frac{2B^{2-t}}{(1-t)^2}, \quad \text{where } \max(\|\boldsymbol{y}\|_{2-t}, \|\hat{\boldsymbol{y}}\|_{2-t}) \leqslant B \tag{11}$$

We here calculate the upper bound using the 1st order Taylor expansion.

$$T_n'(\boldsymbol{w}_{t+1}, \theta_{t+1}) + \frac{M}{n!} \|\boldsymbol{w}_{t+1} - \boldsymbol{w}_t\|^n \simeq \nabla_{\boldsymbol{w}} J(\boldsymbol{w}_t, \theta_{t+1}) \cdot (\boldsymbol{w}_{t+1} - \boldsymbol{w}_t) + (b-a)\|\boldsymbol{w}_{t+1} - \boldsymbol{w}_t\| \tag{12}$$

$$\frac{\partial L_{t_1}^{t_2}}{\partial \boldsymbol{w}_i} = \frac{\partial L}{\partial \hat{y}} \frac{d\hat{y}}{d\hat{a}} \frac{d\hat{a}}{d\boldsymbol{w}_i} \tag{13}$$

$$= \sum_j \frac{\partial}{\partial \hat{y}_j} \left( -\frac{1}{1-t} y_i \hat{y}_i^{1-t} + \frac{1}{2-t_1} \hat{y}_j^{2-t_1} \right) \frac{\partial \hat{y}_j}{\partial \hat{a}_i} \frac{d\hat{a}_i}{d\boldsymbol{w}_i} \tag{14}$$

$$= \sum_j (\hat{y}_j - y_j) \hat{y}_j^{-t_1} \frac{\partial \left( \exp_{t_2}(\hat{a}_j - \xi_{t_2}(\hat{\boldsymbol{a}})) \right)}{\partial \hat{a}_i} \frac{d\hat{a}_i}{d\boldsymbol{w}_i} \tag{15}$$

$$= \sum_j (\hat{y}_j - y_j) \hat{y}_j^{-t_1} \hat{y}_j^{t_2} \left( \delta_{ij} - \frac{\partial \xi_t(\boldsymbol{a})}{\partial a_i} \right) \frac{d\hat{a}_i}{d\boldsymbol{w}_i} \tag{16}$$

$$= \sum_j (\hat{y}_j - y_j) \hat{y}_j^{t_2 - t_1} \left( \delta_{ij} - \frac{\hat{y}_i^{t_2}}{\sum_{j'} \hat{y}_{j'}^{t_2}} \right) \boldsymbol{z} \tag{17}$$

We here calculate the upper bound using the 2nd order Taylor expansion.

$$T_n'(\boldsymbol{w}_{t+1}, \theta_{t+1}) + \frac{M}{n!} \|\boldsymbol{w}_{t+1} - \boldsymbol{w}_t\|^n$$

$$\simeq \nabla_{\boldsymbol{w}} J(\boldsymbol{w}_t, \theta_{t+1}) \cdot (\boldsymbol{w}_{t+1} - \boldsymbol{w}_t) + \frac{1}{2}(\boldsymbol{w}_{t+1} - \boldsymbol{w}_t)^{\mathsf{T}} \nabla_{\boldsymbol{w}}^2 J(\boldsymbol{w}_t, \theta_{t+1})(\boldsymbol{w}_{t+1} - \boldsymbol{w}_t) + \frac{(b-a)}{2} \|\boldsymbol{w}_{t+1} - \boldsymbol{w}_t\|^2 \tag{18}$$

$$\frac{\partial^2 L_{t_1}^{t_2}}{\partial \boldsymbol{w}_i \partial \boldsymbol{w}_i} = \frac{\partial}{\partial \boldsymbol{w}_i} \sum_j (\hat{y}_j - y_j) \hat{y}_j^{t_2 - t_1} \left( \delta_{ij} - \frac{\hat{y}_i^{t_2}}{\sum_{j'} \hat{y}_{j'}^{t_2}} \right) \boldsymbol{z} \tag{19}$$

$$= \sum_j \left\{ \frac{\partial \hat{y}_j}{\partial \hat{a}_i} \frac{\partial \hat{a}_i}{\partial \boldsymbol{w}_i} \hat{y}_j^{t_2 - t_1} \left( \delta_{ij} - \frac{\hat{y}_i^{t_2}}{\sum_{j'} \hat{y}_{j'}^{t_2}} \right) \boldsymbol{z} \right. \tag{20}$$

$$+ (\hat{y}_j - y_j) \frac{\partial \hat{a}_i}{\partial \boldsymbol{w}_i} \frac{\partial \hat{y}_j^{t_2 - t_1}}{\partial \hat{a}_i} \left( \delta_{ij} - \frac{\hat{y}_i^{t_2}}{\sum_{j'} \hat{y}_{j'}^{t_2}} \right) \boldsymbol{z} \tag{21}$$

$$\left. - \frac{\partial}{\partial \hat{a}_i} \frac{\partial \hat{a}_i}{\partial \boldsymbol{w}_i} \frac{\partial \xi_t(\boldsymbol{a})}{\partial a_i} (\hat{y}_j - y_j) \hat{y}_j^{t_2 - t_1} \boldsymbol{z} \right\} \tag{22}$$

$$= \sum_j \left\{ \hat{y}_j^{t_2} \left( \delta_{ij} - \frac{\hat{y}_i^{t_2}}{\sum_{j'} \hat{y}_{j'}^{t_2}} \right) \boldsymbol{z} \hat{y}_j^{t_2 - t_1} \left( \delta_{ij} - \frac{\hat{y}_i^{t_2}}{\sum_{j'} \hat{y}_{j'}^{t_2}} \right) \boldsymbol{z} \right. \tag{23}$$

$$+ (\hat{y}_j - y_j)(t_2 - t_1) \hat{y}_j^{t_2 - t_1 - 1} \hat{y}_j^{t_2} \left( \delta_{ij} - \frac{\hat{y}_i^{t_2}}{\sum_{j'} \hat{y}_{j'}^{t_2}} \right) \boldsymbol{z} \tag{24}$$

$$\left. - \frac{1}{Z_t} \sum_{j'} t \exp_t(a_{j'} - \xi_t(\boldsymbol{a}))^{2t-1} \left( \delta_{ij'} - \frac{\partial \xi_t(\boldsymbol{a})}{\partial a_i} \right) \left( \delta_{jj'} - \frac{\partial \xi_t(\boldsymbol{a})}{\partial a_j} \right) (\hat{y}_j - y_j) \hat{y}_j^{t_2 - t_1} \boldsymbol{z} \right\} \tag{25}$$

### A.1.1 Comparison to the convergence of $\theta$ in ASNG-NAS

We assess criteria for terminating ASNG-NAS. We use CIFAR-10 dataset. Although ASNG-NAS does not explicitly have the stopping scheme, the convergence of $\theta$ can be monitored. We plot it with the proposed upper bound. The olive and green lines in Figure 4 show the convergence of $\theta$ explained in subsection 3.3 and purple and magenta lines are the upper bound $r_t$ used for our stopping criterion. Red and orange lines are the CIFAR-10 test errors calculated by the one-hot architectures with the inherited weights at each epoch. The convergence of $\theta$ "converges" earlier whereas the quantities for our stopping criterion keep decreasing so as the test errors. In general, the acquired one-hot network is retrained after the architecture search. After the convergence of $\theta$ reaching almost 1, the apparent subsequent improvement in the test accuracy is considered as the the learning of weights. If we are going to retrain weights, it is not efficient to wait for the improvement in test accuracy due to learning of weights. In addition, it is well-known that the validation accuracy during architecture search and the test performance after retraining do not correlate. Moreover, the learning rate significantly affects the convergence speed as well as the final test performance[2]. For rapid convergence, the architecture search tends to prefer simple operations which only improve the valid accuracy in the short term. This may not always result in high test performance. Then, practitioners may want to conduct comprehensive assessment of a few criteria. The proposed upper bound $r_t$ is a candidate for such criterion.

We note that there is no significant difference in convergence of $\theta$ and test error between the cases for the bi-tempered logistic loss and the cross entropy loss. Our metric for the cross entropy loss has larger variance particularly at the early stage than the bi-tempered logistic loss, which is considered to be due to the approximation of upper bound as explained in subsection 5.2[3].

## A.2 Additional results

We here provide additional experimental results. We evaluate our stopping criterion on DARTS using NAS benchmarks including NAS-Bench-201, NATS-Bench, NAS-Bench-301, and NAS-Bench-ASR.

---

[2]We choose the configuration to ensure the correlation.

[3]We substitute the loss value at the first epoch into the upper bound $b$ of the cross-entropy loss. This empirical estimation of the upper bound can be loose and introduce variance to the results.

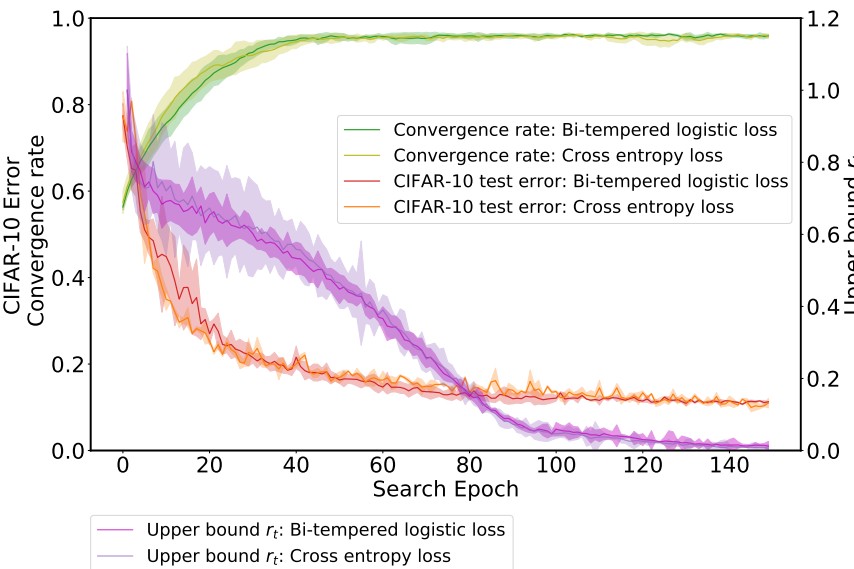

Figure 4: Convergence of $\theta$ equipped with ASNG-NAS, CIFAR-10 test error (included both in one plot for convenience, corresponding to the left axis.), and our criterion.

### A.2.1 Additional results on NAS-Bench-201 and NATS-Bench

The baseline accuracies of the obtained architectures are taken from benchmarks and the quantity used for our stopping criterion are shown in Figure 5. The results are averaged for 5 trials and the solid lines are the mean values and the shaded areas are the standard deviations. We show all of 5 trials with the red, blue, green dashed lines representing the terminated timings by ATNAS, RobustDARTS, and DARTS+. Although the three termination methods have different timings for different trials, RobustDARTS and DARTS+ are successful in stopping before performance degradation possibly caused by domination of skip-connects. The ATNAS stops the search around similar timing or earlier than the other two previous early-stopping strategies. Hence, ATNAS is a strong candidate for the automatic termination method for NAS.

### A.2.2 Additional results on NAS-Bench-301 and NAS-Bench-ASR

We evaluate versatility of our proposed stopping criterion using benchmarks including NAS-Bench-301 (Siems et al., 2020) and NAS-Bench-ASR (Mehrotra et al., 2021). NAS-Bench-301 is a large benchmark which consists of $10^{18}$ architectures and their performances on CIFAR-10. NAS-Bench-ASR is a benchmark for automatic speech recognition tasks. The baseline errors of the obtained architectures are taken from benchmarks (Figure 6). The results are averaged for 5 trials and the solid lines are the mean values and the shaded areas are the standard deviations. Similarly to the NAS-Bench-201 and NATS-Bench, all of the three termination methods are successful in stopping search before performance degradation (Figure 6a). In particular, while DARTS+ and RobustDARTS terminate search where test errors start to increase, ATNAS is able to terminate much earlier at plateaus with lower test errors. For NAS-Bench-ASR, the ATNAS and the RobustDARTS stop the search when the performance saturates (Figure 6a). ATNAS terminates search earlier than RobustDARTS. ATNAS is a versatile automatic termination method for a variety of NAS tasks.

### A.3 A table of final performance

The following tables 1,2,3,4 show the test CIFAR-10/CIFAR-100/ImageNet16 accuracies of the acquired architectures at the terminated search epoch by ATNAS, RobustDARTS, and DARTS+, respectively. The averaged values over 5 trials are shown with the standard deviations.

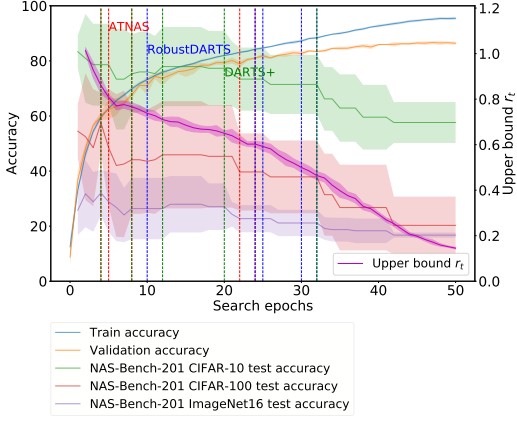

(a) DARTS with bi-tempered logistic loss trained on CIFAR-10 and NAS-Bench-201 baseline accuracy.

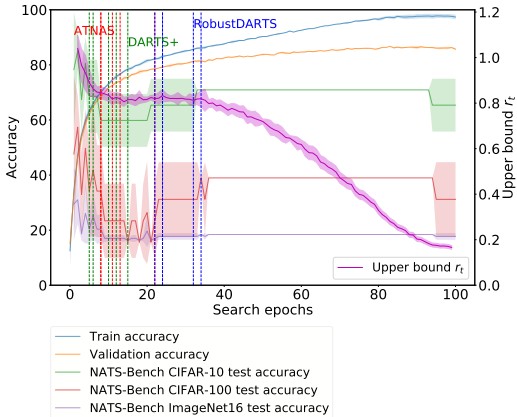

(b) DARTS with bi-tempered logistic loss trained on CIFAR-10 and NATS-Bench baseline accuracy.

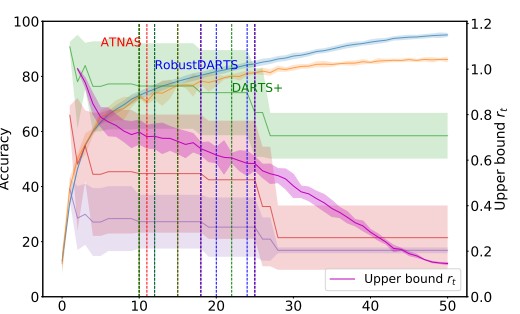

(c) DARTS with cross entropy loss trained on CIFAR-10 and NAS-Bench-201 baseline accuracy.

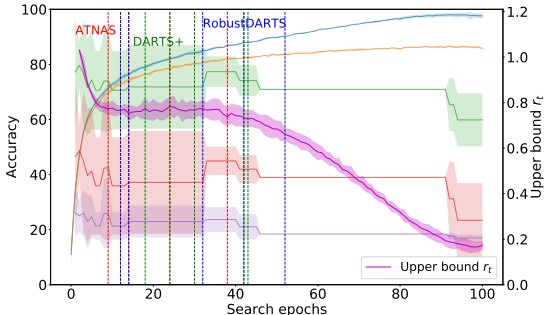

(d) DARTS with cross entropy loss trained on CIFAR-10 and NATS-Bench baseline accuracy.

Figure 5: DARTS with bi-tempered logistic loss and cross-entropy loss trained on CIFAR-10 and NAS-Bench-201 and NATS baselines. All of the termination timings are shown.

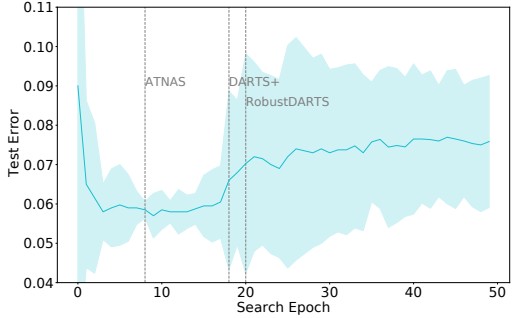

(a) DARTS with bi-tempered logistic loss trained on CIFAR-10 and NAS-Bench-301 baseline test errors are reported. The dashed vertical lines refer to the termination epochs.

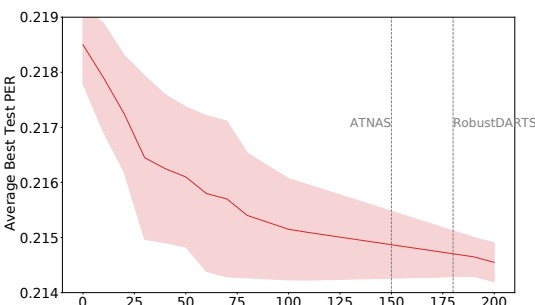

(b) DARTS with bi-tempered logistic loss equipped with NAS-Bench-ASR. The average best test Phoneme error rate (PER) is reported. The dashed vertical lines refer to the termination epochs.

Figure 6: DARTS with bi-tempered logistic loss equipped with NAS-Bench-301 and NAS-Bench-ASR benchmarks.

Table 1: A table of final performance. DARTS with bi-tempered logistic loss trained on CIFAR-10 and NAS-Bench-201 baseline accuracy.

|  | Termination epoch | Test accuracy of CIFAR-10 | CIFAR-100 | ImageNet16 |
| --- | --- | --- | --- | --- |
| ATNAS | $10 \pm 7$ | $86.67 \pm 5.91$ | $54.39 \pm 21.92$ | $31.71 \pm 10.03$ |
| RobustDARTS | $23 \pm 7$ | $82.84 \pm 4.93$ | $23.90 \pm 20.00$ | $25.96 \pm 11.54$ |
| DARTS+ | $22 \pm 11$ | $83.35 \pm 5.69$ | $41.46 \pm 24.61$ | $24.38 \pm 12.01$ |

Table 2: A table of final performance. DARTS with cross entropy loss trained on CIFAR-10 and NAS-Bench-201 baseline accuracy.

|  | Termination epoch | Test accuracy of CIFAR-10 | CIFAR-100 | ImageNet16 |
| --- | --- | --- | --- | --- |
| ATNAS | $16 \pm 6$ | $86.36 \pm 5.49$ | $50.73 \pm 23.21$ | $28.35 \pm 10.03$ |
| RobustDARTS | $22 \pm 3$ | $81.54 \pm 9.31$ | $43.56 \pm 18.75$ | $23.90 \pm 9.81$ |
| DARTS+ | $15 \pm 5$ | $76.61 \pm 13.46$ | $46.65 \pm 21.04$ | $24.80 \pm 11.69$ |

Table 3: A table of final performance. DARTS with bi-tempered logistic loss trained on CIFAR-10 and NATS-Bench baseline accuracy.

|  | Termination epoch | Test accuracy of CIFAR-10 | CIFAR-100 | ImageNet16 |
| --- | --- | --- | --- | --- |
| ATNAS | $17 \pm 8$ | $54.39 \pm 21.92$ | $41.82 \pm 23.76$ | $20.34 \pm 5.26$ |
| RobustDARTS | $28 \pm 5$ | $70.75 \pm 10.08$ | $23.40 \pm 13.49$ | $17.71 \pm 1.21$ |
| DARTS+ | $10 \pm 5$ | $61.15 \pm 15.31$ | $34.17 \pm 16.69$ | $18.68 \pm 2.50$ |

Table 4: A table of final performance. DARTS with cross entropy loss trained on CIFAR-10 and NATS-Bench baseline accuracy.

|  | Termination epoch | Test accuracy of CIFAR-10 | CIFAR-100 | ImageNet16 |
| --- | --- | --- | --- | --- |
| ATNAS | $19 \pm 12$ | $74.93 \pm 13.40$ | $43.96 \pm 21.35$ | $27.43 \pm 10.08$ |
| RobustDARTS | $32 \pm 16$ | $72.10 \pm 10.79$ | $35.37 \pm 13.83$ | $20.58 \pm 4.32$ |
| DARTS+ | $31 \pm 11$ | $70.81 \pm 9.87$ | $37.61 \pm 15.26$ | $19.86 \pm 4.40$ |

Table 5: A table of final performance. DARTS with bi-tempered logistic loss trained on CIFAR-10 and NAS-Bench-301 baseline errors.

|  | Termination epoch | Test errors of CIFAR-10 |
| --- | --- | --- |
| ATNAS | $10 \pm 3$ | $0.0574 \pm 0.0057$ |
| RobustDARTS | $15 \pm 5$ | $0.0585 \pm 0.0039$ |
| DARTS+ | $12 \pm 5$ | $0.0584 \pm 0.0028$ |

Table 6: A table of final performance. NAS-Bench-ASR baseline Average Best Test PER.

|  | Termination epoch | Average Best Test PER |
| --- | --- | --- |
| ATNAS | $148 \pm 19$ | $0.21487 \pm 0.00061$ |
| RobustDARTS | $172 \pm 25$ | $0.21476 \pm 0.00048$ |

## A.4 Computational time

In this subsection we investigate the computational time of the proposed method. In particular we confirm that the computation time for the automatic determination of the termination threshold is not dominant

with respect to that of the architecture search. In the following table we show the computational time of stopping criterion (equation 6), determining the threshold (equation 8) and the architecture search at each epoch. The experimental setup follows the section 5. The search is executed with either ASNG-NAS or DARTS. We train the model with CIFAR-10. The experiments are repeated for five trials and the mean values and standard deviations are reported (Table 7).

Table 7: Computational time for each epoch, architecture search, stopping criterion (equation 6) and termination threshold (equation 8) (in seconds).

|  | Each epoch | Architecture search | Stopping criterion | Threshold |
| --- | --- | --- | --- | --- |
| ASNG-NAS | $87.4143 \pm 5.3360$ | $73.4244 \pm 3.998$ | $8.8542 \pm 0.7148$ | $4.4021 \pm 0.5232$ |
| DARTS | $527.7572 \pm 11.3356$ | $500.7134 \pm 6.8206$ | $19.1664 \pm 2.3588$ | $6.4564 \pm 0.6235$ |
| DARTS+ | $-$ | $-$ | $-$ | N/A |
| RobustDARTS | $-$ | $-$ | $87.5423 \pm 11.4333$ | N/A |

