# OpenReview forum: "ATNAS: Automatic Termination for Neural Architecture Search"
_TMLR — Rejected by TMLR_

### Review · Reviewer_LxMQ · 2022-12-18

**Summary Of Contributions:**

This work proposed a more versatile and principled early-stopping strategy to overcome performance degradations during weight-sharing-based neural architecture search (NAS). The authors estimated the gap of generalization errors of the previous and current search steps with respect to the architecture parameters. This stopping threshold can be automatically determined at each search step without too much extra cost.

**Audience:**

Yes

**Broader Impact Concerns:**

In general, the motivations and methods hold. But I personally believe for the question of "early stopping in DARTS", its impact is a bit limited in the current machine learning community.

**Claims And Evidence:**

Yes

**Requested Changes:**

In addition to the questions above:
1. The term “convergence rate” used in section 3.3 (and also A.1.1) is misleading: https://en.wikipedia.org/wiki/Rate_of_convergence
2. Fig. 1 and Fig. 5: Too many colors and shadow regions, making it hard to read (e.g. which curve is for the accuracy or upper bound).
3. A table of final performance (comparing with state-of-the-art methods) is required.

**Strengths And Weaknesses:**

Strength
1. The paper writing and organization are good.
2. The idea is well-motivated.

Weakness:
1. I first have a clarification question: for the generalization gap you want to measure, is it the generalization of 1) the supernet, or 2) the sampled architecture (at its convergence)?
* For (1), I don’t see why it is not available during the search. You can directly take the supernet’s weights and $\theta$ and run the validation, right?
* For (2), I agree that at each search step, when you sample an architecture from $\theta$, it is infeasible to train this single-path network to converge and get its generalization error. But to my understanding, Eq. 3 is not for a single-path network, but for the supernet.
2. Have you ever tried to directly compare the validation error of the supernet at step $t-1$ and $t$ versus Eq. 8?
3. In Fig. 2 and Fig. 3: is the accuracy of DARTS always keep dropping during the search? In my experience, the accuracy of DARTS should first increase then drop.
4. In Fig. 1, ATNAS early stops when $r_t$ is lower than 0.1, but why does ATNAS stop at 0.8~0.9 in Fig. 3? Or did I misunderstand anything?

---

### Review · Reviewer_R4YT · 2022-12-23

**Summary Of Contributions:**

This work proposes an early stopping criterion to stop NAS early to avoid overfitting. Specifically, such a criterion is based on the evaluation of a gap between expectation values of generalisation errors of the previous and current search steps with respect to the architecture parameters. The experiments in multiple NAS Benchs have shown that this work can reduce the cost of the search process while maintaining high performance.

**Audience:**

Yes

**Broader Impact Concerns:**

No concern about the ethical implications of this work.

**Claims And Evidence:**

Yes

**Requested Changes:**

Please see the weaknesses above.

**Strengths And Weaknesses:**

## Strengths
> + Well motivated: How to make the NAS process more efficient is an important direction and the authors clearly state this motivation
> + Easy to follow: The writing logic flow is easy to follow
> + Well-studied related works: the related works are well studied

##  Weaknesses
> + Novelty: there is already a lot of work on the early stopping criterion in NAS. The proposed one is not strictly proved to be better than those priors works, and the experiments show that in some cases (e.g., Fig 3 (c)), the performance of the proposed one is quite close to the priors works. Thus, I suggest that the authors can spend more effort on explaining why the proposed one can be an option as compared to the previous simple but effective one.
> + Limited experiments: the experiments are all on image classification datasets and only small-scale ones (e.g., CIFAR-10/100 and ImageNet-16). Not sure whether it is still on effective technique in other tasks and larger datasets because we cannot conclude that the proposed one will definitely be better than previous ones from the method section.
> + Missing details of training setup: The training setup at the beginning of the experiments section does not provide sufficient information. I suggest the authors to include more details on their training recipe selections and clarify whether those configurations are consistent with other baselines.

---

### Review · Reviewer_A1KH · 2022-12-27

**Summary Of Contributions:**

The paper propose a new early stopping criterion for neural architecture search. This allow to reduce the computation cost of NAS but can also be used to reduce the overfitting of this approaches.


**Audience:**

Yes

**Broader Impact Concerns:**

No concerns

**Claims And Evidence:**

Yes

**Requested Changes:**

It can be interesting to add the following changes:

- Add large scale experiments on ImageNet-1k

- Add multi-seed experiment to do an overfitting analysis

**Strengths And Weaknesses:**

Strengths:

- The paper is well written and easy to follow.
- The idea is interesting and can be useful in practice.
- The code is provided in supplemental this is a good practice.

Weaknesses:

- Large scale experiments: The results are interesting but large scale experiments are missing, the majority of the results are on CIFAR there is only one evaluation on ImageNet16. It would be interesting to have experiments on large scale dataset like ImageNet-1k. Indeed, it is currently difficult to know if the proposed approach is useful in a large scale context.

- Overfitting analysis: It would be interesting to have a study of overfitting with multi-seed experiments for instance. This can help to understand how the proposed method help to mitigate overfitting. Indeed, this is highlighted in the abstract but there is no analysis of this specific point in the paper.

---

### Author Response · Authors · 2023-01-23
**Update on the manuscript.**

Thank you once again for your careful reading of the manuscript and constructive remarks. We have updated the manuscript with results on the additional experiments. The program codes are also updated.

The updates are as follows:

- A.2.2: Results on NAS-Bench-301 and NAS-Bench-ASR
- A 3: Tables of final performances

For NAS-Bench-301, we have the similar results with NAS-Bench-201/NATS-Bench. We would like to emphasise that while DARTS+ and RobustDARTS terminate search when test errors start to rise, ATNAS is able to stop early at plateaus with lower test errors. For NAS-Bench-ASR, ATNAS is able to reduce search cost by stopping search earlier than robustDARTS.

---

### Decision · Action_Editors · 2023-02-02

**Recommendation:** Reject

**Comment:**

This paper proposes a new early-stopping strategy for neural architecture search (NAS). The reviewers raise some concerns about this paper's contribution and the method's generalization ability. For example, Reviewer A1KH and R4YT both ask experiments on large-scale datasets to investigate the effectiveness. Though authors update the manuscript in the rebuttal phase, they do not answer these important issues, and a reviewer posts a negative score after reading the response. Therefore, I think this paper is not ready to be published and recommend rejecting it.

**Audience:**

Few individuals will be interested in the current version as its effectiveness on large-scale datasets has not been verified.

**Claims And Evidence:**

The method's generalization ability has not been verified well. Especially, It is difficult to know whether this method works well in a large-scale dataset.